# Measurement Systems for Use in the Navigation of the Cannula–Guide Assembly within the Deep Regions of the Bronchial Tree

**DOI:** 10.3390/s23042306

**Published:** 2023-02-19

**Authors:** Tomasz Nabagło, Zbisław Tabor, Piotr Augustyniak

**Affiliations:** Faculty of Electrical Engineering, Automatics, Computer Science and Biomedical Engineering, AGH University of Science and Technology, 30-059 Krakow, Poland

**Keywords:** bronchial diagnostics, medical mechatronics, virtual bronchoscopy, image-guided intervention

## Abstract

Background: The purpose of this paper is to present the spatial navigation system prototype for localizing the distal tip of the cannula–guide assembly. This assembly is shifted through the channel of a bronchoscope, which is fixed in relation to the patient. The navigation is carried out in the bronchial tree, based on maneuvers of the aforementioned assembly. Methods: The system consists of three devices mounted on the guide handle and at the entrance to the bronchoscope working channel. The devices record the following values: cannula displacement, rotation of the guide handle, and displacement of the handle ring associated with the bending of the distal tip of the guide. Results: In laboratory experiments, we demonstrate that the cannula displacement can be monitored with an accuracy of 2 mm, and the angles of rotation and bending of the guide tip with an accuracy of 10 and 20 degrees, respectively, which outperforms the accuracy of currently used methods of bronchoscopy support. Conclusions: This accuracy is crucial to ensure that we collect the material for histopathological examination from a precisely defined place. It makes it possible to reach cancer cells at their very early stage.

## 1. Introduction

Bronchoscopy is aimed at diagnosing changes such as cancer tissue within the bronchial tree. This endoscopic procedure is usually applied in the case of malignant lung lesions [1,2] in order to provide samples to microscope histopathologic examination. The bronchoscope is a tool equipped with a flexible probe including a working channel, a light source and an optical wire that allows observation of the interior of the bronchial tree. Unfortunately, image-based navigation with a bronchoscope is possible up to approximately the third carina (branching level) of the bronchial tree. The diameter of the bronchoscope distal end is about 6 mm, and the narrowing branches of the bronchial tree reach this diameter at the third carina. The remaining distance to the end of the branches of the bronchial tree is approx. 100 mm. This distance can be traveled only by the cannula–guide assembly introduced into the bronchial tree through the working channel of the bronchoscope. The assembly is much thinner; however, it does not have the ability to transmit an image.

Nowadays, two methods for supporting the navigation of the cannula–guide assembly to the peripheral parts of a bronchial tree are currently used in clinical practice [3]. The first method is based on fluoroscopy [4,5], which has, however, well-recognized limitations, primarily the ambiguity of directly localizing the tip of a catheter in a volume of a respiratory tract based on 2D fluoroscopic images. The second solution uses electromagnetic navigation bronchoscopy (ENB) [6,7,8] coupled to virtual bronchoscopy. ENB systems continuously generate electromagnetic fields sensed by a special sensor mounted on a tip of a guiding device. It should be remembered that sometimes reaching the terminal branches of the bronchial tree, the diameters of which can reach even 1–2 mm, enables early detection of a cancer. 

Due to the high price of the ENB solution as well as the extensive experience required by the fluoroscopy-based method, we decided to develop a cheaper solution that simplifies the deep bronchoscopy procedure. Due to the intuitive optical navigation, which however is limited to an area not exceeding the third carina, we considered using virtual reality to extend the scope of bronchoscopy in the solution proposed by us. The analysis of the maneuvers performed within the previously known bronchial tree by the cannula–guide assembly can unambiguously determine the position of the tip of the guide within this tree, which we will demonstrate later in the paper. The demonstrated system is a prototype of the navigation system of sensor-assisted bronchoscopy (SAB).

In this type of navigation system, the virtual model of the patient’s bronchial tree is needed. To construct a virtual model of the patient’s bronchial tree, a conventional CT has to be performed first. This virtual model is usually used in the ENB method; moreover, it may be used in the fluoroscopy-supported method in order to navigate the area behind the third carina. In the case of bronchoscopy under fluoroscopic guidance with CT, due to the fact that the physician can observe in real time the image of the moving guiding device in the real bronchial tree, the error in the location of the nodule is negligible from the point of view of the procedure effectiveness.

However, they are not comparable because ENB is a 3D method as opposed to the fluoroscopy-assisted method, which is 2D method. In the fluoroscopy-assisted method, the important negative aspect is the patient’s exposure to X-rays during the procedure [5]. The physician performing the procedure is also exposed to this radiation. In the case of the ENB solution the patient is also exposed to X-rays, because the conventional CT is needed to construct a virtual model of the patient’s bronchial tree,. Similarly, in the solution proposed in this study, which estimates the position of the tip of the guide in the bronchial tree based on information obtained from three sensors. Nevertheless, the difference is related to the fact that in the case of conventional CT, maximum exposure time is 12 s at a dose of 474 µSv, and it is associated with CT for the virtual bronchial tree model, which is used in all three methods. Because in the fluoroscopy-supported method, the treatment is carried out under conditions of exposure to X-rays, the operator must wear a special outfit and observe special safety procedures. Furthermore, for the patient, testing under these conditions is associated with additional anxiety. In this procedure, in the worst case, when the lesion size is below 10 mm, the patient X-ray absorption is approx. 925 µSv, whereas the physician X-ray absorption is approx. 93.5 µSv [5].

Another recently proposed method of bronchoscopy procedure is called Shape-Sensing Robotic-Assisted Bronchoscopy (SSRAB) [9,10,11]. From the point of view of the robotics discipline, the system is based on the robotic manipulator, which is controlled by the operator. This person operates the bronchoscope, which is integrated with the manipulator. The diameter of this bronchoscope is approx. 3.5 mm. After reaching the point of the bronchial tree, where the diameter of a branch of the bronchial tree is too narrow to continue shifting the bronchoscope, the operator can introduce the cannula–guide assembly to penetrate the dipper part of the bronchial tree. This operation is usually performed with the assistance of fluoroscopy. For this reason, we cannot compare the SSRAB method with our concept because it is related to cannula–guide assembly navigation; therefore, its performance area starts exactly at the location where the bronchoscope is stopped by narrowing branch of the bronchial tree. Therefore, the system concept proposed by us may support the SSRAB method rather than competing with it.

Moreover, solving the problem of accurate non-image navigation will allow the probe to reach cancer cells at a very early stage of development, enable prompt initiation and improve the effectiveness of therapy.

### Options of the Optical Navigation

The possibilities of image-based navigation are being explored by several leading research groups. Michalski et al. [12] show that cannula displacement can be measured in an automated manner by analyzing video sequences recorded by the video camera of the bronchoscope. They observed that the accuracy of the cannula displacement measurement is subject to an error for which the size depends on the velocity of the cannula displacement. Bułat et al. [13] and Twardowski et al. [14] described a bronchoscope-positioning support system during biopsy in the area of the bronchial tree. The system identifies the location in the bronchial tree on the basis of images from the endoscopic camera and artificial images from a virtual model of the bronchial tree based on the patient’s computed tomography (CT) data. A similar problem was undertaken by Chien et al. [15]. However, as a tool to match real bronchoscopy images with virtual bronchoscopy images, they used a position-tracking method using current frames from the bronchoscope and verifying the position of the real bronchoscope image with the image extracted from the 3D model using an adaptive neural network system based on fuzzy reasoning. These systems can help physicians to navigate effectively in the area where they can use the bronchoscope vision, and, therefore, only to the third carina of the bronchial tree. Nevertheless, by knowing the three-dimensional image of the bronchial tree from the tomography and analyzing the maneuvers made with the guide, as well as the path traveled by the cannula–guide set, it is possible to precisely predict the position of the guide tip and even visualize its displacement in real time [16]. All clinically available solutions supporting navigation in peripheral bronchoscopy are based on methods of direct location of the distal tip of the guide inside the airways. This leads to solutions with limited functionality, such as in the above-mentioned bronchoscope, or solutions that are complex and expensive, as in the case of ENB. On the other hand, it can be assumed that monitoring and collecting data on manual maneuvering of a handle mounted at the nearer end of a guide can provide sufficient information to accomplish the task of estimating the position, orientation, and bend of the distal end of the guide.

This kind of solution may be applied for documentation and verification whether diagnostic procedures during classic bronchoscopy were conducted properly.It may make it possible to record maneuvers realized by physicians, and based on these records, it may make it possible to find faults in their maneuvers or even in generally accepted procedures. This system may evaluate the correctness of the examination. It can also contribute to improving the quality of the physician’s work; furthermore, it can generate data for simulators used to train physicians. In reference to this potential of the system, the papers of Vieira et al. [17] and Kennedy et al. [18] were analyzed. They describe an overview of bronchoscopy simulators, as well as their application and impact on medical education. They present the prospects of using these types of simulators, which, due to the dangers of learning on patients, make it possible to prepare physicians for this type of procedure in a safe virtual environment. Khare et al. [19] proposed a non-contact virtual bronchoscopy navigation system for planning and conducting bronchoscopy. A human clinical trial has shown that with this system, the bronchoscope-to-lesion time is often nearly 3 min shorter than the guide times of other technician-assisted systems. In such systems, the analysis of the maneuvers performed could reduce this time even more. 

Gibbs et al. [20] proposed a system that incorporates an automatic optimal route-planning method in bronchoscopy examination, which integrates known route constraints. The system offers a natural translation of the multidetector-computed tomography-based route plan into the live guidance strategy via data fusion of tomographic and bronchoscopy-generated visual data. Cornish et al. [21], similarly to us, used an optical sensor; however, they used it to determine the linear displacement and rotation of the bronchoscope. Based on the information from the sensor, they determine the position of the bronchoscope tip inside the bronchial tree. A similar task was undertaken by Michalski et al. [22]; however, the position of the bronchoscope was determined in this case on the basis of the shape of the bronchial tree and the maneuvers performed by the tip of the bronchoscope. Observation of these maneuvers is based on data from two IMUs (inertial measurement unit) BNO055 sensors.

## 2. Materials and Methods

To test the hypothesis that the displacement, rotation and bending of the distal tip of the guide can be predicted from manual maneuver records with a guide handle mounted at the proximal end of the guide, three devices were designed and prototyped: device D1 to control cannula displacement; device D2 to control the angle of rotation of the handle; and device D3 to control the displacement of the handle ring, determining the bending of the guide tip (Figure 1).

The red line represents the route to the desired destination. Manual maneuvers with the guide handle are reflected in the corresponding animation of the virtual tip. Moving the cannula moves the virtual tip into the virtual bronchoscopy space. 

The whole system which encloses devices D1, D2 and D3 communicates throughout a virtual serial port available via Bluetooth. Measured values of the cannula–guide assembly shift, guide rotation angle and the guide-tip bending angle are transmitted to the PC with a visualization system of the virtual bronchoscopy (see Figure 2).

### 2.1. Recording the Cannula Displacement

The first D1 device records and transmits information about the displacement of the cannula-guide assembly. This device is placed in the proximal hole of the working channel of the bronchoscope (Figure 1). In this device, we decided to apply the ADNS-9800 High-Performance Laser Stream Gaming Sensor (Avago Technologies Ltd., San Jose, CA, USA), which makes it possible to control the displacement of the tracked surface in two perpendicular directions. It works on the basis of laser-light reflection analogous to sensors commonly used in optical mice. However, unlike the sensor used in a computer mouse, the ADNS-9800 uses laser light and can, therefore, accurately track the movements of very smooth surfaces. The sensor captures successive images of the slide guard surface. A digital signal processor (DSP), embedded in the sensor, processes images, calculates the values of relative displacements in two perpendicular directions and, thus, determines the direction and magnitude of the displacement of the sampled surface. Because laser light is used in the mentioned solution, the movements of seemingly smooth surfaces such as surfaces and cannulas can be successfully tracked. The relative displacements of the test surface are directly available in the memory registers of the ADNS-9800 sensor and transmitted to an external microprocessor, where they are collected to calculate the total displacement of the cannula in each of the two probed directions: axial and transverse movements (rotations). Furthermore, we are interested only in displacements in the axial direction, because the cannula is not turned around its longitudinal axis during peripheral bronchoscopy (i.e., only the guide turns around). The ADNS-9800 sensor communicates with an external microcontroller (Arduino technology) via SPI (Serial Peripheral Interface). This type of communication interface is fast enough to transmit information about the cannula displacement to an external tracking system. The Arduino microcontroller is used to acquire data about the cannula displacement, which is then transmitted wirelessly to a PC using Bluetooth 4.0 technology.

The D1 device consists of specially designed housing unit enclosing an electronic system that collects data on the cannula displacement. Details of the housing design are shown in Figure 3. The housing unit consists of a bottom part (Figure 3a) and a cover (Figure 3b). Inside the bottom part, there is the lower cannula bed, and inside the cover, there is the upper cannula bed. The two beds, when assembled together, form a cylindrical guide channel with a radius equal to the outer radius of the cannula. The axis of the guide channel formed by the two beds is parallel to the longitudinal axis of the translation measurement of the ADNS-9800 sensor. 

The ADNS-9800 is attached to the upper housing bed in such a way that its distance from the bed can be adjusted using four knobs. Proper adjustment of the distance from the sensor to the upper bed prevents the production of artifacts by the presence of a guide inside the cannula. Because the cannula surface is composed of a partially transparent material, moving the guide can affect cannula movement measurements. In addition, the housing unit has a neodymium magnet attached to the lower bed to position the guide (composed of ferromagnetic steel) at the bottom of the cannula channel (Figure 4). The Arduino board is placed in the lower part of the housing unit, below the lower cannula bed (see Figure 3c).

In the process of operation, the D1 device is placed on the proximal end of the working channel of the bronchoscope using a clamp located on the case (Figure 3c). Then, the lid of the box is opened. The tip of the cannula is inserted into the working channel of the bronchoscope through the device guide channel, and the cannula is placed between the beds in the guide channel. Then, the cover is closed. When the measurement system is activated, data on cannula displacements are transmitted to a computer, where they can be read from any serial port terminal.

A working prototype of the D1 device is presented in Figure 5. In Figure 5a, the lid, which is opened, enables the introduction of the canula–guide assembly to the guiding channel of the bronchoscope. Figure 5b presents ready-to-work device D1, and Figure 5c presents a scheme of its electronic circuit, which includes its data acquisition unit (Arduino Micro), laser sensor ADNS-9800, wireless communication unit HM-11 and a power supply with a lithium polymer rechargeable battery.

The housing units of all of the devices were produced via Fused Deposition Modeling (FDM) technology. Due to the high resistance to external conditions and the appropriate mechanical properties, acrylonitrile butadiene styrene terpolymer (ABS) material was used to print the housing units. This material is also biologically inert in the solid state.

The device communicates with a PC containing software visualizing the movement of the tip of the guide in a virtual model of the bronchial tree via Bluetooth. When Bluetooth devices were paired, they formed a standard serial communication port.

### 2.2. Recording the Rotation Angle of the Guide Handle

The D2 device is based on an inertial measurement unit (IMU) sensor mounted inside a specially designed housing unit. In our prototype, we used the BNO055 sensor, a 9-axis absolute orientation sensor (Bosch Sensortec GmbH, Kusterdingen, Germany). The BNO055 IMU is equipped with a 3-axial magnetometer, gyroscope and accelerometer. In addition, it has a signal processor that processes data from three sensors. If required, it returns the values of three angles—roll, jaw, and pitch—in its memory registers. The BNO055 sensor communicates with an external Arduino microcontroller via the I2C interface. This interface type was arbitrarily chosen by the manufacturer. 

The D2 device measures and transmits the value of the rotation angle of the guide handle. The D2 device housing is mechanically attached to the guide handle. This housing unit with the IMU sensor inside is designed such that the axis associated with roll angle is parallel to the axis of the guide (Figure 6). The value of this angle is returned in the BNO055 memory register. In the case of the D2 device, whenever it is switched on, an associated microcontroller Arduino sends previously prepared calibration data to the IMU. Thanks to this mechanism, the above angles are measured in relation to the previously designated absolute position of the device.

For convenience (mainly to minimize the size of the device), we used an Arduino micro board. The use of I2C requires only two wires to transmit data on the angle rotation of the guide handle. The Arduino microcontroller then transmits information about the rotation angle of the guide handle to a PC using Bluetooth 4.0.

In the design, we used a housing unit consisting of three elements: the lower chamber containing the D2 with BNO055; the middle chamber containing the D3 device, which is described in the next chapter; and the upper chamber containing the Arduino microcontroller (Figure 7a). The chamber containing the BNO055 has a special clamp and pin for attaching the housing unit to the guide handle. The housing stem pin fits into the socket in the guide handle and provides a rigid connection between the two structures. With this housing design, the axis of the chamber is parallel to the axis of the guide. Inside the chamber, there is a rectangular recess of the size corresponding to the dimensions of the BNO055 and the edges parallel to the chamber walls. In this way, the roll angle measured by the BNO055 corresponds to the axial rotation angle of the guide.

For comparison, in Figure 7b, the working prototype of the assembled devices D2 and D3 is presented. Its electronic circuit is presented in Figure 8. 

These are the same electrical supply elements as in the D1 device case. The same data acquisition unit (Arduino Micro) cooperates with sensors. The IMU sensor (BNO055) is the main element of the rotation angle measurement system of the guide (D2). An assembly of a Trinket 5V controller and a linear potentiometer is the bend-angle measurement system of the guide tip (D3). The controller, with its A/D converter, measures resistance of the potentiometer and sends its value throughout the I2C bus to the Arduino Micro. The same bus is used for communication between the Arduino and the BNO055 sensor. The pair of measurements from both of the sensors are sent wirelessly throughout virtual serial ports with usage of the HM-11 element.

### 2.3. Reporting on Deflection of the Distal End of the Guide

The device D3 captures the deflection of the flexible end of the guide in relation to the displacement of the guide ring connected to the guide tip with a steel wire. The deflection control device of the flexible end of the guide is based on a displacement sensor located in the guide handle (Figure 7).

The concept of the D3 is based on the design of the guide handle. The handle has a special ring that can move proximally and distally from a certain neutral position in the axial direction (Figure 9). Depending on whether the ring moves proximally or distally, the tip of the guide bends in two opposite directions. Thus, it can be assumed that the proper calibration and the distance to which the ring is moved, as well as the detection of the direction of the actual value of shift from the neutral position, are sufficient to determine the deflection angle of the guide tip.

The deflection angle of the guide tip varies in the range [−45°, 105°], resulting from the mechanical design of the guide. We have determined that the distal displacement of the ring at a distance of 3.2 mm from the neutral position results in a deflection angle of −45°. When the ring is moved proximally to the distance of 2.6 mm from the neutral position, it results in a deflection angle of 105°. Thus, the linear piece calibration function was adopted, as shown in the Figure 10.

The calibration function was constructed for the OLYMPUS CC-6DR-1 guide device model cooperating with the OLYMPUS BF TE2 Bronchoscope.

In the prototype, we used a linear potentiometer with total resistance 10 kΩ. The resistance of the potentiometer is read by a small electronic unit via the analog input and sent to the Arduino microcontroller and then transmitted wirelessly (Bluetooth) to the PC. Signals from each device were synchronized by the PC application of the Virtual Bronchoscopy. This application repeatedly sends a signal to each device in a 10 ms interval to request for an update of the actual position of the distal end of the cannula–guide assembly on the PC screen. In response to this signal, each device sends actual measurement data to the PC.

## 3. Results

The three presented measurement systems cooperate with each other in order to precisely estimate the maneuvers performed during the movement of the cannula–guide assembly in the deep regions of the bronchial tree. Although they work within one positioning system, in order to verify their measurement accuracy, they should be treated as separate subsystems. To verify the accuracy of the D2 and D3 systems, appropriate reference systems were used. These reference systems will be briefly referred to later as R2 and R3.

### 3.1. Linear Displacement Measurement Accuracy 

In order to test the accuracy of the cannula displacement measurement, four series of measurements were performed. In each series, the cannula was gradually inserted into the working channel of the bronchoscope at a distance of up to 100 mm. Every 10 mm, readings from the reference system and from the D1 device were recorded. The differences between the readings D1 and the reference system are shown in the Figure 11.

In Figure 11, one can see that the error of measurement D1 relative to the reference frame does not exceed 2 mm, and in most cases, it remains below 1 mm (average: −0.18, std.: 0.65). Readings of the D1 device correlate very strongly with reference measurements (correlation coefficient greater than 0.9998). We also examined the hysteresis of the measuring system. The displacement of the cannula was measured by the D1 device and the reference system during subsequent insertions and removals of the cannula from the working channel of the bronchoscope. The hysteresis is shown in Figure 12. 

In the last experiment, the cannula was alternately inserted and removed from the working channel to a total displacement distance of 800 mm. In the first series, the cannula was inserted at a distance 200 mm, then pulled out at a distance 200 mm, and this combination of movements was repeated twice. In the second series, the cannula was inserted at a distance 100 mm, then pulled out at a distance 100 mm, and this combination of movements was repeated four times. Each experiment was repeated 30 times. Discrepancies between the readings of the D1 device and the reference measurements were recorded. In the first experiment, the mean value and the deviation value were 0.88 and 0.56 mm (maximum 2.29 mm and minimum 0 mm). In the second experiment, the mean and standard deviation were 0.71 and 1.09 mm (maximum 2.86 mm and minimum −1.43 mm).

### 3.2. Angular Displacement Measurement Accuracy 

Reference measurements of the roll angle of the guide were performed with two reference systems. The first reference system R1 was applied to provide reference measurements of the roll angle of the guide handle. The second device R2 was applied to provide reference measurements of the roll angle of the tip of the guide.

To check the measurement accuracy of the axial rotation angle of the guide handle, four series of experiments were carried out. In each series, the handle rotated gradually from the set starting position. Approximately every 10 degrees, the rotation angle was read from the reference system and from the D2 device. The handle was rotated up to an angle of 360 degrees from the initial position. Before proceeding with further analysis, a constant value was subtracted from the D2 readings because the zero position D2 did not always correspond to the zero position of the absolute encoder that was an element of the reference system R1. The angular differences between the readings D2 and the reference system are shown in Figure 13. It can be seen that the error of measurement D2 relative to the reference frame does not exceed 1.5 degrees. It was found that the D2 readings strongly correlated with reference measurements (correlation coefficient greater than 0.99995). No hysteresis was recorded in the D2 measurement system in relation to R1.

Because the guide is composed of flexible wire, one can expect the axial rotation angle of the guide handle only in some ranges corresponding to the axial rotation angle of the guide tip. To check how the rotation angle of the axial guide tip depends on the axial rotation angle of the guide handle, four series of measurements were performed. At the beginning of each series, the readings D2 and R2 were calibrated such that both systems showed zero angle of rotation (with no discrepancy between D2 and R2). Then, in each series, the handle was rotated from the starting position. Depending on the measuring series, the axial rotation angle of the handle was increased or decreased by 10 to 30 degrees. In each step, readings from the R2 reference system and with the D2 device were recorded. In each series, the axial rotation angle of the handle increased from 0 to 180 degrees, then decreased to −180 degrees, and then increased to 0 degrees. Thanks to this sequence of measurements, hysteresis was recorded in the D2 measuring system. The results of the measurements are shown in Figure 14.

It is clear that the difference between the axial rotation angle of the handle and the axial rotation angle of the guide tip can be up to 60 degrees, which can be a problem in the clinical application of the proposed method. The shape of the hysteresis curve suggests methods that can be used to control the angle of rotation of the tip with much better accuracy. Indeed, Figure 14 shows that after a change in the direction of rotation (for example, by approximately 180 degrees or −180 degrees), the discrepancy between the R2 and D2 readings decreases. Thus, it can be assumed that if the tip of the guide device is to be rotated from a neutral position to a certain angle α, it must be rotated first by a larger one, for example π/2 + α, and then, it should rotate to an angle α in the direction of the neutral position. To test this hypothesis, four series of measurements were performed. The target axial rotation angle of the tip was 30, 60, 90, 120, 150 and 180 degrees clockwise (twice) and counterclockwise (also twice). The handle was rotated by a target angle of plus 90 degrees, and then rotated to the target angle towards the neutral position. When the handle was rotated to the target angle, readings D2 and R2 were recorded, and the difference between the two devices was calculated. The results of this experiment are shown in Figure 15. As it was demonstrated, for the proposed mode of operation, the error of the axial rotation angle of the guide tip relative to the axial rotation angle of the handle can be significantly reduced.

### 3.3. Measurement Accuracy of the Guide Tip Bending Angle 

In order to assess the accuracy of the deflection angle measurement of the guide tip at the beginning, the reference system R3 was calibrated against the protractor. This reference system is based on an electronic compass system that was statically fixed to the ground. Above the compass was the guide tip with a strong neodymium magnet rigidly attached to its last segment. Its precision was verified by its supplier. We tested the entire bending angle range from −45° to 105°. The difference between the value found by the automatic reference system and the value set on the protractor was less than 1 degree; thus, we treat this reference system as highly accurate. The correlation coefficient between the automated reference system and visual measurements of the tip deflection angle was 0.9994.

Taking into account the above calibration results of the automatic reference measurement system of the guide tip deflection angle, in the next experiments, we compared the readings D3 with the readings from the automated reference system R3. To check the correctness of the measurement of the deflection angle of the guide tip, four series of measurements were carried out. In each series, the tip was gradually bent from the minimum to the maximum value. The values of the deflection angle, measured by device D3 and the reference measurement system R3, were recorded and compared. The angular differences between the D3 readings and the reference system are shown in Figure 16.

The error of measurement D3 reported to the reference system can be up to 20 degrees. It was found that there is a relationship between error values and reference values of the deflection angle (correlation coefficient from 0.82 to 0.91). It can also be said that D3 measurements strongly correlate with reference measurements (correlation coefficient greater than 0.985). We also investigated hysteresis in the measuring system. The hysteresis curve is shown in Figure 17.

## 4. Discussion

The developed mechatronic system is directly coupled with the basic tools used during peripheral bronchoscopy—with the cannula and with the guide. For this reason, no modifications to the standard peripheral bronchoscopy procedure are needed for the potential clinical use of the designed devices. The physician will continue to use standard equipment, but—in addition—his maneuvers will be recorded, analyzed and possibly used to assist navigation to the cancerous lesion. In addition, the measurement process does not constitute a mechanical load for the doctor performing a standard bronchoscopy procedure, nor does it limit the range of maneuvers during this procedure. Because the developed mechatronic system is based on standard sensors, the costs associated with the implementation of the system are very low in relation to currently used spatial navigation systems in the bronchial tree.

The novelty of this solution is the use of three independent measuring devices placed on the proximal part of the cannula–guide assembly in order to estimate the position of the guide tip on the basis of the compilation of the devices measurement results.

The proposed system was developed not to replace but to assist the navigation to the destination in peripheral bronchoscopy. In particular, similarly to electromagnetic navigation, the recorded data can be processed with software used in peripheral bronchoscopy. To this point, before the actual bronchoscopy, the patient undergoes a computed tomography (CT) scan, which is used to generate a personalized model of airways also known as ‘virtual bronchoscopy’. CT scans and virtual bronchoscopy are used to plan the navigation path to the target (cancerous lesion) from an original location accessible to the bronchoscope under video guidance. After the bronchoscope is inserted into its initial position, the cannula and guide are inserted together into the working channel. Then, after the tip of the guide slides out of the distal end of the working channel, the physician is given the opportunity to observe the animation of the virtual guide tip in the space of the virtual bronchoscopy. The data representing the manual maneuvering of the guide handle, which are recorded by the proposed prototype mechatronic system, serve to properly animate the virtual guide tip and appropriately reposition the virtual guide tip in the space of the virtual bronchoscopy (Figure 1c). For example, when a cannula is inserted into the working channel, the virtual bronchoscopy view is moved deeper and deeper into the virtual airways. The actual distance the virtual camera will move in virtual airway light can be calculated from the recorded cannula shift and the pixel size of the CT image. This animation, together with the path to the target displayed in the virtual space of the bronchoscopy, constitutes a complete system to support navigation in peripheral bronchoscopy.

The accuracy of the above-described system can be compared with the accuracy of electromagnetically navigated bronchoscopy. Hautmann et al. [23] assessed the accuracy of electromagnetic navigation during bronchoscopy. They used multi-planar reconstruction of a CT data set for 3D localization of the catheter. The sensor position generated in the navigation system was monitored by fluoroscopy. During this operation, the corresponding errors of distance were measured. The mean error of distance between the sensor tip position and the fluoroscope-verified reference position were 10.4 mm in the lateral position and 12.5 mm in the apical position. The mean error of distance between the sensor tip and two endobronchial registration points at the end of the procedure were 4.2 mm in the lateral and 5.1 mm in the apical position. Eberhardt et al. [24] described a mean navigation error of electromagnetically navigated bronchoscopy equal to 9 ± 6 mm (range, 1 to 31 mm). Makris et al. [25] reported that the accuracy of the navigation process, as expressed by the average of CT-to-body divergence, was 4 ± 0.15 mm, whereas the distance between the sensor probe and the center of the target lesion before each attempt for biopsy was 8.7 ± 0.8 mm. Luo [26] described electromagnetically navigated bronchoscopy with a new marker-free recording method that uses bronchial tree midlines and a bronchoscope-attached medium to which an electromagnetic signal is attached to align preoperative images and electromagnetic systems. After application of this method, the average fiducial registration error was reduced from 8.7 to 6.6 mm, and the average target registration error, which indicates all tracked or navigated bronchoscope position accuracy, was reduced from 6.8 to 4.5 mm compared to the default registration methods.

In the final version of the system, correction of the respiratory movements influence should also be applied. During respiratory movements, the bronchoscope, as well as the tip of the guide of the cannula–guide assembly, moves slightly forward and backward in the bronchial airways. Because of this movement of the real elements, the position of the virtual tip of the guide in the virtual lumen of the bronchial tree must be adequately corrected. This problem can be solved with well-known 2D to 3D registration methods described by Higgins et al. as well as Helferty et al. [27,28].

## 5. Conclusions

Summarizing the analysis of the presented solution, we compared it with competitive solutions in this article. Its advantage in terms of its measurement accuracy should also be emphasized. The measurement error of the D1 device relative to the reference system did not exceed 2 mm, which shows its advantage over the so-called electromagnetic navigation, which has been used for years in medicine, in which the displacement measurement error reaches 9 mm [24]. For device D2, the maximum error was 10°; for device D3, the maximum error was 20°. However, due to deviations between adjacent branches of the bronchial tree above 30°, these errors are acceptable for the correct navigation process. Moreover, because the direction of movement is known, based on the calibrated repeatable hysteresis value, assuming a specific linear accuracy, it is possible to increase the precision of determining the current value of the guide tip rotation angle as well as the guide tip deflection angle.

To highlight the advantages of the described solution, we have presented them in Table 1 in comparison with two competing solutions used currently in bronchoscopy.

For more interested readers, the authors present a collection of drawings, circuit diagrams, CAD models and source codes in the Appendix A.

## Figures and Tables

**Figure 1 sensors-23-02306-f001:**
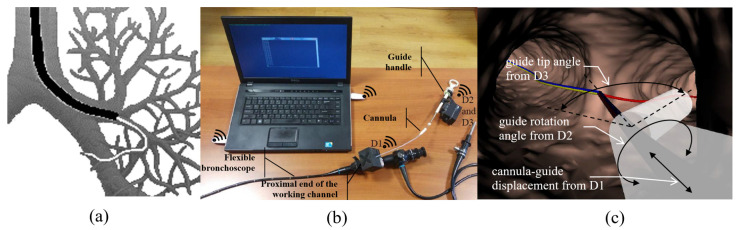
The problem of navigation inside the bronchial tree and its solution. (**a**) The problem of maneuvering the guide in order to reach the designated place in the deep part of the bronchial tree. (**b**) Full measurement system prototype. (**c**) Virtual animated guide tip (transparent white element) in the virtual bronchoscopy space.

**Figure 2 sensors-23-02306-f002:**
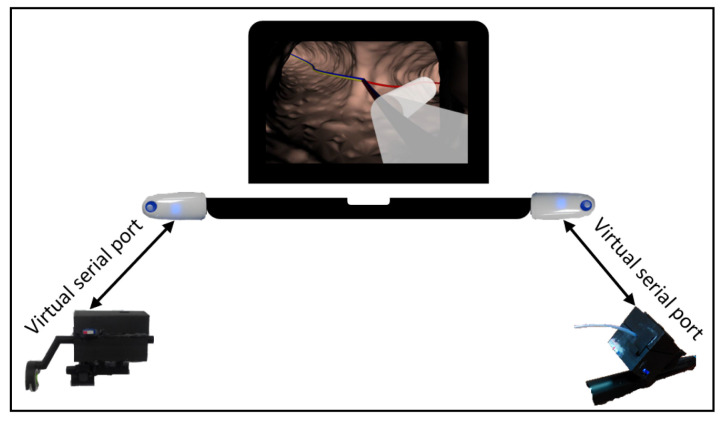
Communication between Devices D1 (on the right), D2 and D3 (on the left) and the virtual bronchoscopy visualization system.

**Figure 3 sensors-23-02306-f003:**
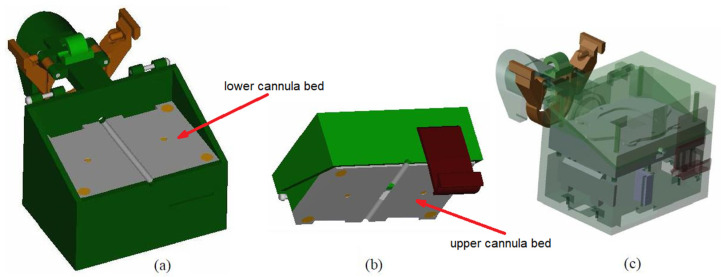
Details of the design of the housing unit of the D1 cannula displacement device: (**a**) the lower part, (**b**) the cover and (**c**) the assembly details. The grey parts in (**a**,**b**) are the lower and upper beds of the cannula. In the beds, both halves of the cannula guide channel are visible.

**Figure 4 sensors-23-02306-f004:**
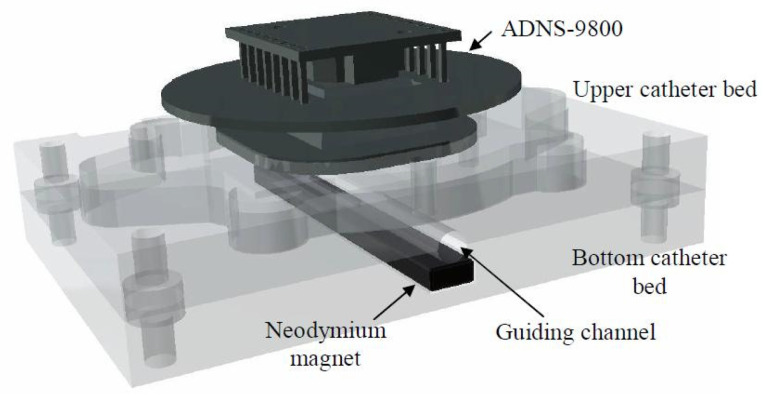
Details of the construction of cannula beds.

**Figure 5 sensors-23-02306-f005:**
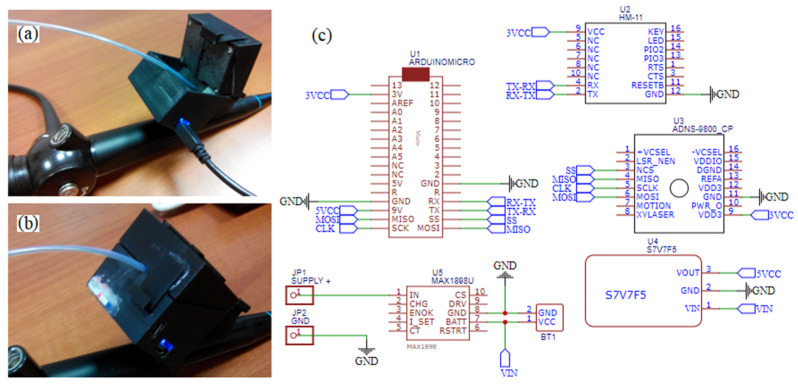
Prototype of the measurement system of the cannula–guide assembly linear displacement: (**a**) insertion of the cannula–guide assembly; (**b**) the prototype in their working condition; (**c**) cordless measurement system circuit.

**Figure 6 sensors-23-02306-f006:**
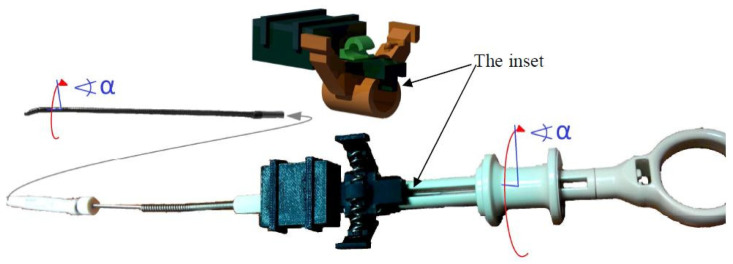
The concept of measuring the rotation angle of the guide handle based on the IMU type sensor.

**Figure 7 sensors-23-02306-f007:**
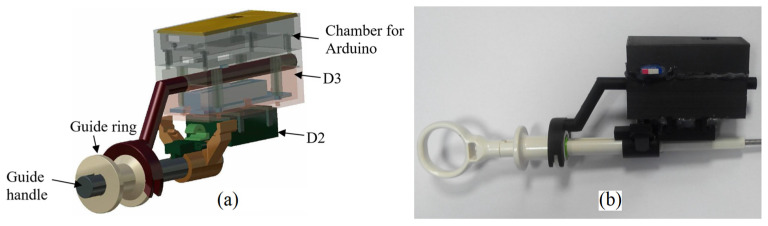
The guide handle rotation angle measurement device and the deflection of the guide tip. (**a**) Details of the housing unit. (**b**) The prototypes in their working condition.

**Figure 8 sensors-23-02306-f008:**
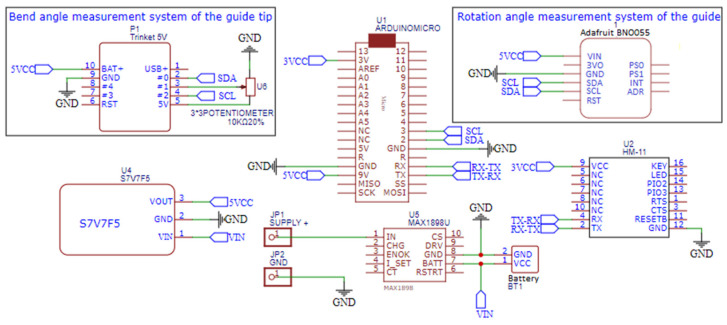
Cordless measurement system circuit of the guide handle rotation angle and the deflection of the guide tip.

**Figure 9 sensors-23-02306-f009:**
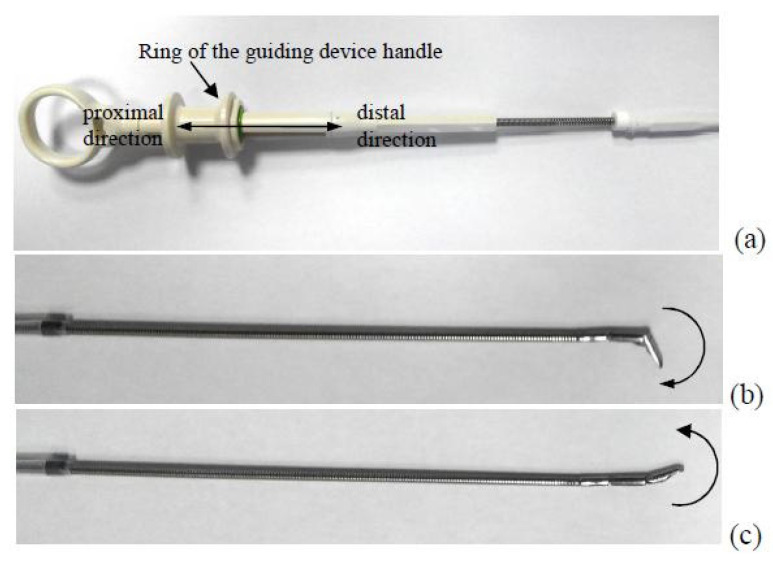
The concept of measuring the deflection of the guide based on the measurement of the displacement of the ring: (**a**) the guide holder ring; (**b**) the tip deflection corresponding to the proximal displacement of the ring; (**c**) the tip deflection corresponding to the distal displacement of the ring.

**Figure 10 sensors-23-02306-f010:**
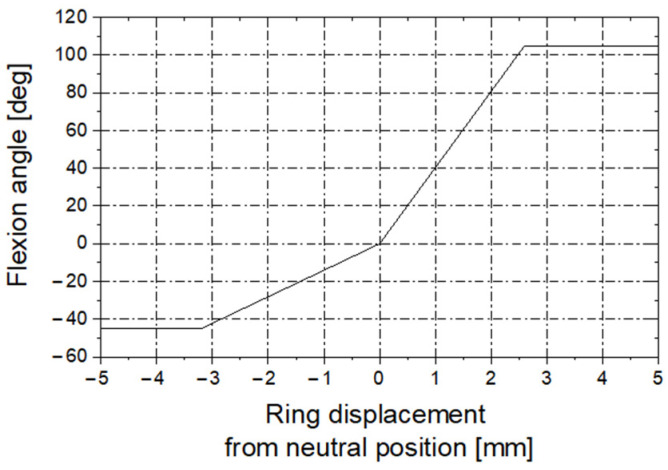
Calibration function: ring displacement vs. deflection of the guide tip.

**Figure 11 sensors-23-02306-f011:**
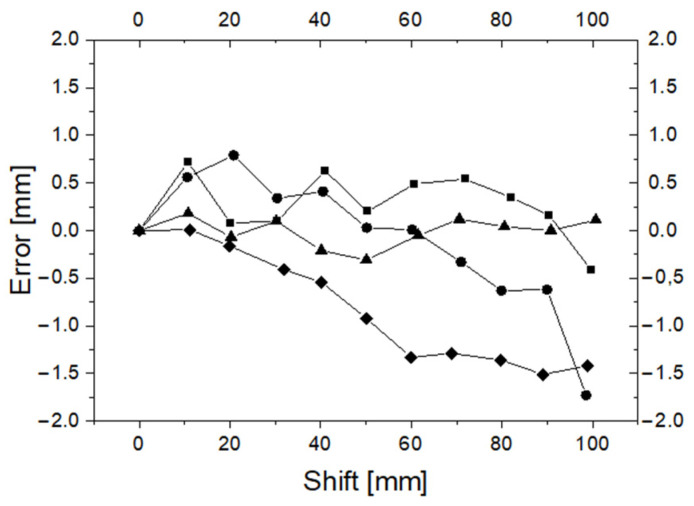
Differences (error) between D1 and reference frame readings. The different symbols correspond to four measurement series.

**Figure 12 sensors-23-02306-f012:**
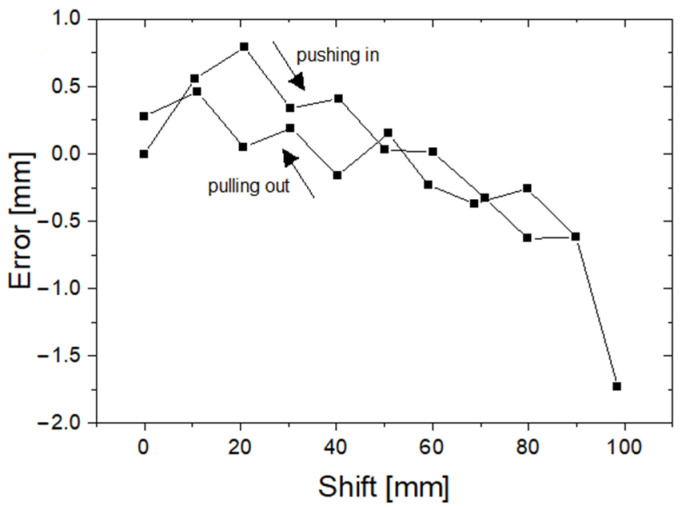
Hysteresis in the measuring system of the D1 device.

**Figure 13 sensors-23-02306-f013:**
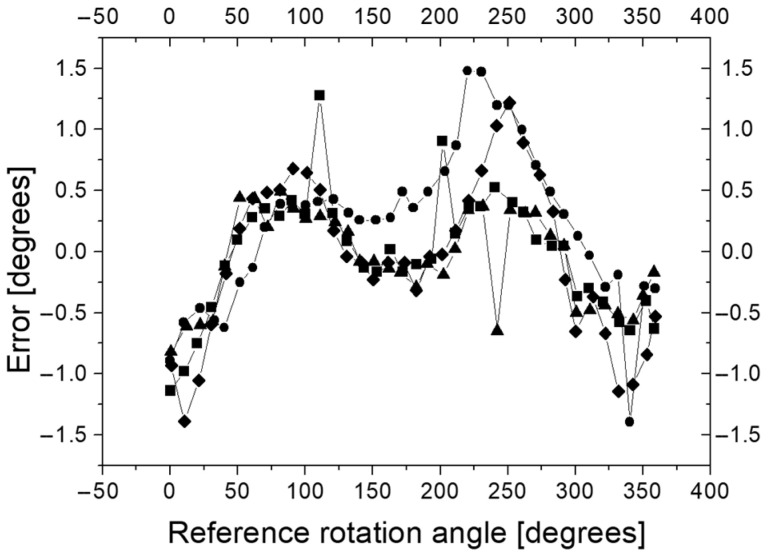
Angular differences (error) between readings D2 and reference frame R1. The different symbols correspond to four series of experiment.

**Figure 14 sensors-23-02306-f014:**
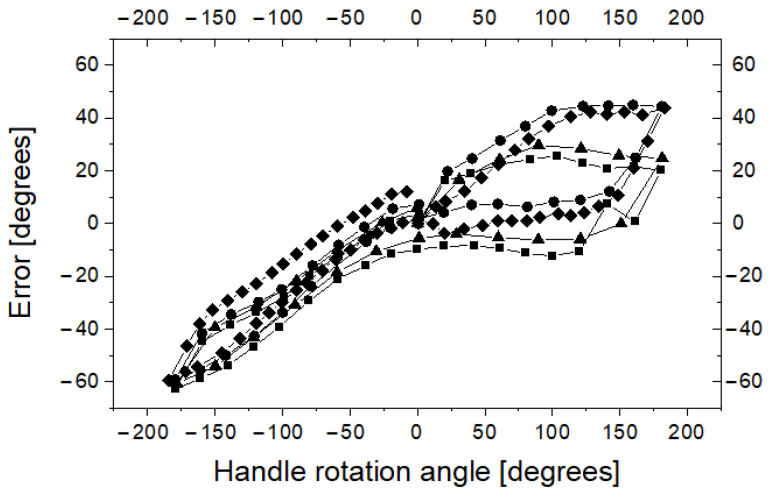
Hysteresis in the measuring system of the D2 device in relation to R2.

**Figure 15 sensors-23-02306-f015:**
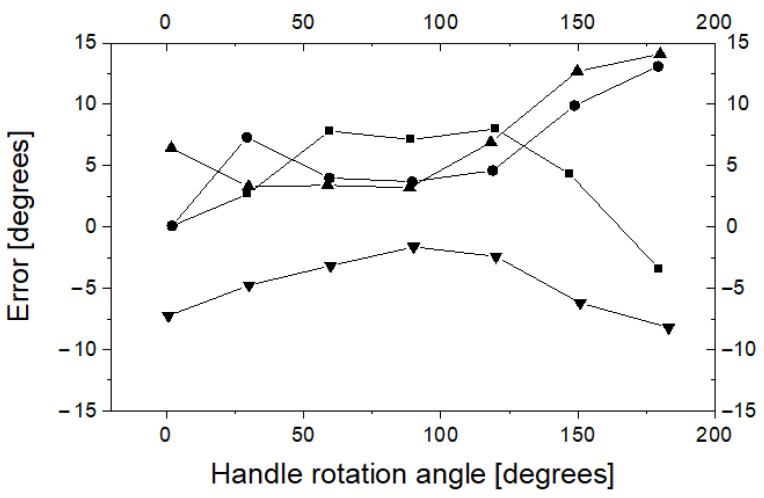
Angular differences (error) between readings D2 and reference frame R2. The different symbols correspond to four series of measurements.

**Figure 16 sensors-23-02306-f016:**
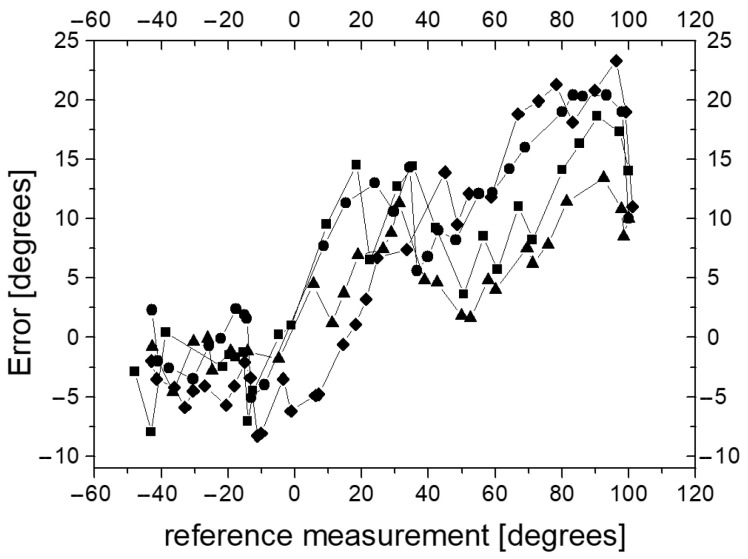
Angular differences (error) between D3 and reference frame readings. The different symbols correspond to four series of measurements.

**Figure 17 sensors-23-02306-f017:**
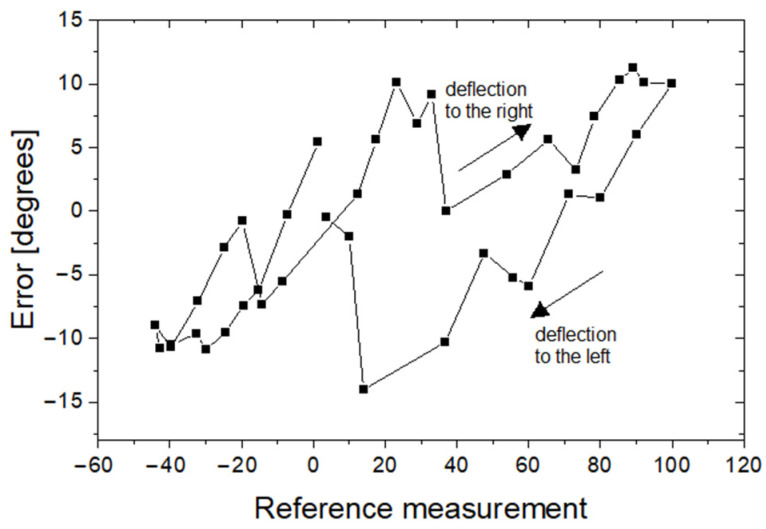
Hysteresis in the measuring system of the angle of deflection of the guide end.

**Table 1 sensors-23-02306-t001:** Comparison of features of competing solutions.

Feature	ENB	FAB	SAB (This Work)
Maximum location error	9 mm	- *	2 mm
X-ray absorption during conventional CT [4]	474 µSv	474 µSv	474 µSv
Patient X-ray absorption during procedure **	0 µSv	approx. 925 µSv	0 µSv
Physician X-ray absorption during procedure **	0 µSv	approx. 93.5 µSv	0 µSv

ENB—electromagnetic navigation bronchoscopy, FAB—fluoroscopy-assisted bronchoscopy, SAB—sensor-assisted bronchoscopy. * Due to the fact that the ENB and SAB methods are three-dimensional methods and the FGB method is a two-dimensional method, the first two are not comparable with the third one in terms of errors resulting from the accuracy of the nodule location in the real bronchial tree. ** This radiation exposure was measured in the case of a procedure in which the lesion size is less than 10 mm [4].

## Data Availability

The data presented in this study are available in the Appendix A.

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
