# Peer review of "Measurement Systems for Use in the Navigation of the Cannula–Guide Assembly within the Deep Regions of the Bronchial Tree"

_sensors, 2023, doi:10.3390/s23042306_

Round 1
Reviewer 1 Report
This paper presented a spatial navigation system for localizing the distal tip of the cannula-guide assembly, which was meaningful for practical bronchial diagnosis. From my opinion, a revision is needed. Some comments are listed as below:
1. In the introduction part, the authors should show the reason that they did this work more directly.
2. Some figures (Figure 9&14) should be reproduced because the different lines had the same symbols.
3. Please add more discussion about the mechanism novelty of your measurement system.
4. Maybe a table could be given to show the comparation between your work and other reports.
5. Please add more relevant references.
6. There are some speaking errors in the text. Please check the language carefully.
Reviewer 2 Report
Title: Measurement systems for use in the navigation of the cannula-guide assembly within the deep regions of the bronchial tree
In this manuscript, Nabagło et al. have proposed a spatial navigation system for localizing the distal tip of the cannula-guide assembly. This assembly is shifted through the channel of the bronchoscope, which is fixed concerning the patient. Overall, the topic is interesting and the study is well performed, however, numerous concerns need to be addressed and the authors are advised to address them all. Without these details, the manuscript is too vague to understand and has no potential for researchers/readers to reproduce the results or replicate the experimentations.
1. The language of the manuscript needs to be reviewed by a native language expert, as there are various grammatical and sentence structuring issues.
2. The abstract should contain information on background, objectives, methods, results and concluding remarks.
3. The introduction lacks a discussion on state-of-the-art methods available. Also, the information has quite inconsistencies and the introduction is not linked properly.
4. Details of the model and its parameters are missing. Such details are crucial for the reproducibility of the method.
5. Methods are reporting the details of prototypes rather than the details of hyperparameters.
6. Results should be updated accordingly, after a thorough comparison.
7. Representations are too naïve. The authors should add some better illustrations as compared to the current ones.
8. I would recommend the author address all these issues, as, without addressing them, the manuscript lacks substantial information.
Round 2
Reviewer 2 Report
Issues raised in the previous review seems to have been adequately addressed.